# Avirulins, a Novel Class of HIV-1 Reverse Transcriptase Inhibitors Effective in the Female Reproductive Tract Mucosa

**DOI:** 10.3390/v11050408

**Published:** 2019-05-01

**Authors:** Michelle D. Cherne, Jesse Hall, Alisha Kellner, Christine F. Chong, Amy L. Cole, Alexander M. Cole

**Affiliations:** Burnett School of Biomedical Sciences, College of Medicine, University of Central Florida, Orlando, FL 32816, USA; michelle_cherne@knights.ucf.edu (M.D.C.); jdhall.01@knights.ucf.edu (J.H.); alishakellner@knights.ucf.edu (A.K.); christinefchong@gmail.com (C.F.C.); amy.cole@ucf.edu (A.L.C.)

**Keywords:** HIV, drug discovery, antimicrobial, mucosa

## Abstract

While extensive research efforts have decreased human immunodeficiency virus (HIV) transmissions and mortalities, new challenges have arisen in the fight to eradicate HIV. Drug resistance to antiretroviral therapy threatens infected individuals, while the prevalence of heterosexual transmission creates an urgent need for therapies effective in the female reproductive tract (FRT) mucosa. We screened a library of 2095 small molecule compounds comprising a unique chemical space, purchased from Asinex Corporation, for antiviral activity against human immunodeficiency virus type 1 (HIV-1) strain BaL and identified several molecular representatives of a unique class of HIV-1 inhibitors, which we termed “Avirulins.” We determined that Avirulins were active against clinical isolates of HIV-1 from genetically variant subtypes, several of which have reduced sensitivity to other antivirals. Avirulins displayed specific dose-dependent inhibition of the HIV-1 drug target, reverse transcriptase (RT). Avirulins were effective against several nucleoside RT-inhibitor resistant strains of HIV-1, as well as one nonnucleoside RT-inhibitor resistant strain containing a 106A mutation, suggesting a noncompetitive mechanism of action. Drugs, which are damaging to the FRT, can increase the risk of HIV-1 transmission. We therefore explored the cytotoxicity of Avirulins against epithelial cells derived from the FRT and found no significant toxicity, even at the highest concentrations tested. Importantly, Avirulin antiviral activity was not diminished in human cervico–vaginal fluid, suggesting retained potency in the milieu of the FRT. Based on these promising results, Avirulins should be valuable chemical scaffolds for development into next-generation treatments and preventatives that target HIV-1.

## 1. Introduction

Human immunodeficiency virus (HIV), the causative agent of acquired immune deficiency syndrome (AIDS), destroys the host immune system by invasion and subsequent destruction of CD4+ lymphocytes, leading to death by opportunistic infections [1,2,3]. At the onset of the HIV/AIDS crisis, patients succumbed to these AIDS related complications within 10 years of infection [4]. The development of HIV-1 antiretroviral therapies (ART) vastly reduced AIDS related complications and mortality and has changed HIV infection from a fatal to chronic disease for those with access to treatment [5]. Presently, 36.7 million people are living with human immunodeficiency virus type 1 (HIV-1), and this number is rising, as new infection rates outpace AIDS-related mortalities [6]. Long-term users of ART are more susceptible to drug resistance [7], and the use of ART creates a subsequent increase in transmitted drug resistance mutations, giving newly infected patients fewer treatment options [7]. This has created the urgent demand for novel therapeutics and preventatives to combat drug resistance and prevent sexual transmission.

A hallmark of HIV infection is establishment of a latent population of infected T cells containing the integrated provirus [8]. This latent stage protects the inactive virus from antivirals, making development of a cure presently unobtainable [8]. Because of this, development of preventative treatments such as vaccines or antivirals that target early stages of infection, are currently considered the most effective strategy for global HIV eradication [8,9]. Development of an effective HIV vaccine has remained elusive, due in part to the rapidly mutating HIV-1 envelope protein, as well as the failure of the adaptive immune system to produce effective neutralizing antibodies following immunization [10,11]. Encouragingly, pre-exposure prophylaxis (PrEP), has shown success in limiting new infections, particularly between serodiscordant men who have sex with men (MSM) [12]. The oral PrEP, Truvada decreased the risk of HIV infection in MSM by up to 90%, while male-to-female transmission was reduced by 75% [12].

Globally, women comprise one third of newly infected individuals, and in sub-Saharan Africa, where the HIV epidemic is most severe, approximately 60% of new infections are in women [6]. Therefore, preclinical development of novel antivirals targeted to the female reproductive tract (FRT) should account for its unique biology. The toxicity of antimicrobials can counteract antiviral benefits by causing damage to the FRT epithelial layer, enabling easier transmission of HIV to underlying target immune cells below [13]. For example, the spermicidal Nonoxynol-9 was a promising topical antimicrobial candidate for prevention of sexually transmitted HIV-1 to women, but results from a clinical trial determined it increased viral transmission, due to toxicity to the FRT epithelium [14]. Antiretrovirals must also retain potency in the biological milieu of the FRT, where high protein concentration, presence of mucus, and dramatic fluctuations in pH may disrupt drug activity [15,16,17].

The success of antiviral therapies and preventatives relies on a broad selection of drug classes for viral suppression without rapid acquisition of drug resistance [18,19,20]. There are currently six approved classes of antiretroviral drugs, targeting five components of the HIV replication cycle: nucleoside reverse transcriptase inhibitors (NRTIs), nonnucleoside reverse transcriptase inhibitors (NNRTIs), integrase inhibitors, protease inhibitors, and entry inhibitors. Reverse transcriptase (RT) inhibitors block conversion of the viral single stranded RNA to DNA after the virion enters target, CD4+ lymphocytes [21]. There are two distinct classes of RT-inhibitors: Competitive NRTIs and noncompetitive NNRTIs. Integrase inhibitors block the insertion of viral DNA into the host genome, while protease inhibitors prevent the HIV protease from assembling new virions in the host cell [22,23]. Some HIV-1 viral fusion inhibitors, which prevent entry into the target cell, have also been approved. Maraviroc, for instance, binds the host cell CCR5 chemokine receptor, blocking its use by the HIV-1 envelope protein gp120 as a coreceptor for attachment [24]. Currently, highly active antiretroviral therapy (HAART) uses a combination of three drugs, usually two NRTIs and either an NNRTI, protease or integrase inhibitor in treating virally naïve patients [25]. The most common regimen for newly infected patients is the NRTIs Tenofovir and Lamivudine in combination with the NNRTI efavirenz [26]. Truvada, the most common PrEP currently in use, utilizes the combination of two NRTIs, Tenofovir and Emtricitabine [27].

More HIV-1 infected patients are receiving ART than ever before, and patients now face lifelong, daily antiviral regimens, increasing the likelihood of drug resistance. For the continued success of ART, novel compounds must be available for combination therapies to suppress viral load and progression to AIDS, ideally without inducing drug-resistant mutants or by combating currently resistant strains. Toward this goal, we screened a large compound library which exists in a unique chemical space, purchased from the drug discovery service, Asinex Corporation, and discovered a compound with substantial anti-HIV-1 activity. We used this compound for a comprehensive in silico compound database search and identified a novel family of 32 anti-HIV-1 compounds that share a phenoxyethyl–pyrazole–pyrrolidine chemical scaffold, which we termed ‘Avirulins.’ Four of these compounds exhibited 50% inhibitory concentrations (IC_50_) at low micromolar concentrations (1.5–3 µM). Avirulins were unique in structure compared to current HIV-1 antivirals and were not cytotoxic at concentrations up to 100 µM. In this report, we investigated the HIV-1 antiviral properties of Avirulins in a range of physiologically relevant conditions, determined their mechanism of action, and assessed their potential application as components of ART or as preventative microbicides in the FRT.

## 2. Materials and Methods

### 2.1. Cell Lines and Viruses

The immortalized cell lines used for HIV-1 infections, TZM-bl (Dr. John C. Kappes, Dr. Xiaoyun Wu, and Tranzyme, Research Triangle Park, NC, USA), and PM1 (Dr. Marvin Reitz), the HIV-1 laboratory adapted strain BaL, the patient isolates 92UG037 (R5, A), 92US712 (R5, B), 93MW960 (R5, C), 93UG067 (R5 × 4, D) 93BR029 (R5, F), and the RT-inhibitor resistant strains HIV-1_74V_/MT-2, Nevirapine-Resistant HIV-1 (N119), HIV-1IIIB A17 Variant, HIV-1RTMF/MT-2, HIV-1_RTMDR1_/MT-2 were acquired from the AIDS Research and Reference Reagent Program (Division of AIDS, NIAID, NIH, Germantown, MD, USA).

Both TZM-bl and PM1 cell lines express the HIV-1 co-receptors CD4, CCR5, and CXCR4 for productive infection [28,29]. TZM-bl contains the HIV-1 tat luciferase reporter construct, which produces luciferase in response to productive HIV infection, making it useful for quantifying infection. TZM-bl was maintained in Dulbecco’s modified Eagle’s medium (DMEM) containing 10% (*v*/*v*) fetal bovine serum (FBS), penicillin, and streptomycin (‘D10’). PM1s are an immortalized pro-myelocytic cell line and were maintained in RPMI 1640 with 10 mM Hepes buffer, streptomycin, penicillin, and 20% (*v*/*v*) FBS (‘R20’). Viruses were propagated in PM1 cells for 6–12 days, the supernatant containing virions was collected, centrifuged, and filtered through a 0.45 micron syringe for removal of cellular debris, then aliquoted and frozen at −80 °C for future infections. The amount of virus in supernatants was quantified using enzyme-linked immunosorbent assay (ELISA) for the HIV-1 envelope protein p24 (Perkin Elmer, Waltham, MA, USA). The viral infectious titer was determined by infection of TZM-bl and with serial dilutions of virus containing supernatants. Effective dilution of viral supernatant was determined as the dilution resulting in infection of 50% of culture cell wells.

Drug resistance of the RTI-resistant viral strains was validated by TZM-bl luciferase infection assay with the RT-inhibitors AZT and Nevirapine, and subsequent comparison to previously published results of drug sensitives [30]. The vaginal, ectocervical, and endocervical epithelium derived cell lines VK2, Ect1, and End1, respectively, were purchased from American Type Culture Collection (ATCC, Manassas, VA, USA) and maintained with keratinocyte serum free media (KSFM) supplemented with bovine pituitary extract, epidermal growth factor, and calcium chloride in tissue culture treated plates, as per ATCC instructions.

### 2.2. Collection of Human Specimens

Peripheral blood mononuclear cells (PBMCs) were collected as follows. Venous blood was drawn from adult volunteers who provided written consent in occurrence with a University of Central Florida Institutional Review Board approved protocol. Blood was drawn into acid citrate dextrose vacutainers (Becton, Dickinson and Company, Franklin Lakes, NJ, USA). PBMCs were separated within an hour of donation via dilution with Dulbecco’s phosphate-buffered saline (DPBS) followed by overlay in lymphocyte separation media (LSM, MP Biomedicals, Santa Ana, CA, USA) and centrifugation at 400× *g* for 30 min. PBMCs were isolated by this density gradient, washed twice with DPBS then resuspended into RPMI containing 10% FBS (R10) and frozen in liquid nitrogen. For treatments, cells were thawed and maintained in R10 in tissue culture treated plates. Infections were performed using PBMCs from two different donors.

Vaginal fluid was collected from postmenarcheal, premenopausal healthy female donors who provided informed consent following the University of Central Florida Institutional Review Board approved guidelines. Donors with current or recent vaginal infections or antibiotic treatments were excluded from collection by a questionnaire. An Instead SoftCup (Ultrafem, New York, NY, USA) was used to collect vaginal fluid by insertion into the vagina for 30 min, per manufacturers instructions, then removed and centrifuged at 1000× *g* for 10 min in a sterile 50 mL conical tube. Collected vaginal fluid was sonicated via ten 2–3 s. pulses using a microtip ultrasound probe. Sonicated vaginal fluid was aliquoted and stored at −20 °C.

### 2.3. TZM-bl Luciferase Infection Assay

TZM-bl were plated at 7000 cells/well in black tissue culture treated 96 well plates, then at ~70% confluency, cells were infected with virus and treated with compound or vehicle. For the initial compound library screening, cells were treated in triplicate with 50 µL of compound diluted in D10 to a final concentration of 50 µM or equivalent vehicle, and 50 µL of 6 ng/mL BaL, as determined by p24 ELISA and viral infectious titer using TZM-bl luciferase assay. All compounds were purchased from Asinex Corporation. Inhibition was determined by comparison to luminescence of the corresponding vehicle control. Infections using all clinical isolates and all RT-inhibitor resistant strains, except HIV-1IIIB A17, were performed with 4 ug/mL of the cationic polymer diethylaminoethyl (DEAE)-dextran for productive infection. After 24 h incubation at 37 °C, 5% CO_2_, treatments were removed, and cells were lysed as instructed for luciferase assay (Bright Glo luciferase system, Promega, Madison, WI, USA). Luciferase was measured using a LMax luminometer (Molecular Devices Corp., Sunnyvale, CA, USA).

### 2.4. In Silico Compound Screening

After the initial screening, the active Avirulin compound was used for a chemical structure database search of the CAS REGISTRY^SM^ using a chemical search program, SciFinder. Thirty-one compounds with 80–95% structural similarity were then purchased from Asinex and screened for antiviral activity using the previously described luciferase assay.

### 2.5. PM1 and PBMC Infection and p24

PM1 cells (3 × 10^6^/mL) were incubated with Avirulin or equivalent vehicle and HIV-1 in RPMI with 2% serum (‘R2’) for 90 min. at 37 °C, 5% CO_2_ at a volume of 100 µL, then diluted with fresh R2 and spun at 200× *g* for 5 min. Cells were then resuspended in 500 µL fresh R20 with equivalent concentration of treatment. Infection with clinical isolates required 2 µg/mL polybrene, a cationic polymer that increase efficiency of retroviral infection. On days 3 and 6 post infection, supernatants were collected, live and dead cell number was determined using trypan blue staining. On day 3 post infection, cells were resuspended in fresh media and diluted to the initial cell concentration and retreated with drug or vehicle. Viral inhibition was determined by measuring concentration of p24 in cell supernatant per million live cells. p24 was detected by HIV-1 p24 ELISA (Perkin Elmer, Waltham, MA, USA).

### 2.6. Cytotoxicity and Cell Viability

FRT cells or TZM-bl were plated at 10,000 cells/well, or 7000 cells/well respectively, in black tissue culture treated 96 well plates and grown to ~70% confluence, then treated with Avirulins or equivalent DMSO vehicle diluted in cell media from a 10 mM stock. After 24 h, cytotoxicity was measured as per instructions of the CytotoxGlo assay (Promega), which used a substrate cleaved to produce luminescence by proteases released in supernatant after cell death. Cell number was normalized to observed cell death by measurement of luminescence produced after total cell lysis of the corresponding well. For PM1 and PBMC infections, cell viability was monitored by trypan blue staining.

### 2.7. Reverse Transcriptase Inhibition Assay

HIV-1 RT inhibition was measured with the Roche Colorimetric Reverse Transcriptase Assay (Roche Diagnostics, Indianapolis, IN, USA). Inhibition of HIV-1 RT was determined by comparison to the potent NNRTI, 5-chloro-3-(phenylsulfonyl)indole-2-carboxamide (CSIC) [31], and the equivalent DMSO vehicle.

### 2.8. Avirulin Antiviral Activity in Clarified Vaginal Fluid

Cervico–vaginal fluid (CVF) was centrifuged at 10,000× *g* for 5 min to clarify mucus. Clarified CVF from five different specimens was pooled then diluted 1:10 in D10 and adjusted to neutral pH using 12.5% NaOH to prevent cytotoxicity to cells. 1-[(3*S*)-3-{5-[2-(2-methylphenoxy)ethyl]-1*H*-pyrazol-3-yl}pyrrolidin-1-yl](Av-5) or vehicle control was added from a 100 mM stock to diluted CVF at 10 µM and incubated at 37 °C in 5% CO_2_ for 2 h. Av-5 (10 µM) in D10 alone were incubated in tandem for consistency. CVF treatments were then diluted further with D10, and 50 µL of each treatment was added to TZM-bl. Fresh D10 (50 µL) containing BaL at 12 ng/mL was then added. After 24 h incubation, the treatment was removed, and cells were lysed and processed for the luciferase assay. Infection inhibition was determined using the luciferase assay described previously.

## 3. Results

### 3.1. Antiviral and Structural Characteristics of Avirulins

A library of 2095 small-molecule compounds obtained from Asinex Corporation were screened for antiviral activity against HIV-1 BaL at 50 µM using the luciferase-based reporter HeLa-derived cell line, TZM-bl [28]. From this library, we discovered one antiviral compound which inhibited 50% of viral infection at 1.5 µM, which was referred to as Av-5. This compound (1-[(3*S*)-3-{5-[2-(2-methylphenoxy)ethyl]-1*H*-pyrazol-3-yl}pyrrolidin-1-yl]) has a novel structure compared to known antiviral HIV-1 compounds. Using this initial hit, we performed an extensive in silico database search for similar structures using the chemical search program, SciFinder, of the Chemical Abstract System (CAS) REGISTRY^SM^, which contains more than 100 million substances [32]. From these results, we selected 31 additional compounds that shared 80–95% structural similarity to the initial antiviral compound for a secondary antiviral screening. The full list and characteristics of these compounds are available in Appendix A. Compounds with detectable viral inhibition at 50 µM were tested using a wider range of concentrations from 1.5 to 50 µM (Appendix A). Avirulin compounds with negligible activity at 50 µM or that were severely cytotoxic based on remaining cell count at 24 h, were not included in further analysis beyond these initial experiments.

Twenty-four out of 32 Avirulin compounds screened had detectable antiviral activity, and all share a common phenoxyethyl–pyrazole–pyrrolidine scaffold. Substitutions at the phenoxyethyl position (referred to in Figure 1 as ‘substituent X’) and nitrogen of the pyrrolidine (referred to as ‘substituent A’ in Figure 1) alter antiviral activity. Figure 1 depicts Avirulins with strong antiviral activity and a representative selection of Avirulins with no antiviral activity for comparison. The structure and activity of all Avirulins are reported in Appendix A.

Highly active Avirulin compounds shared the Avirulin scaffold and a ketone attached to substituent A. Av-5, Av-14, and Av-26 had a strong electronegative group at the ortho position of the Avirulin phenoxy ethyl group (substituent position X_1_) and a bulky or electronegative group attached to a ketone, terminating in a methyl group at carbon 2 of the pyrazole (‘substituent A’). The highly active Av-27 deviates from the other three lead compounds, with a cyclohexane at the substituent A position, and the strong electronegative group meta to the phenoxyethyl position (position X_2_) rather than ortho (position X_1_). High activity Avirulin compounds shared the same chemical space, but deviated in functional group addition, suggesting further chemical modifications may improve anti-HIV-1 activity.

Four moderately active Avirulins (Av-11, Av-28, Av-19, and Av-8) (Appendix A) with activities ranging from IC_50_ 3.6–15 µM, all share the ketone present in high activity compounds, but have bulkier groups attached at substituent A. Certain moderately active compounds have additional methyl groups, or methyl groups located at different positions on the substituent A pyrazole than located on the highly active compounds Av-5 and Av-14. Av-11 was similar in activity to Av-14 at 3.6 µM IC_50_, but was not included as a lead compound for further analysis due to cytotoxicity observed above 25 µM. Moderately active compounds also deviated at the phenoxyethyl portion of the scaffold, with strongly electronegative groups meta and para, respectively, rather than at the ortho position. Six out of 16 minimally active Avirulins did not contain a ketone group at the cyclopentane of the Avirulin scaffold (Appendix A). The remaining minimally active compounds contained this ketone (Appendix A), but each had the electronegative group at meta and para positions, likely causing the observed reduction in antiviral activity. Avirulin compounds with no activity, such as Av-15 and Av-23, tended to have larger and less electronegative groups at substituent X.

The four most active Avirulins, Av-5, Av-14, Av-26, and Av-27, were chosen as lead compounds for further analysis. Assessment of anti-HIV-1 activity of the lead compounds was repeated using a wider range of concentrations ranging from 0.78–100 µM (Table 1, Figure 2a). Av-5 was the most active, Av-26 and Av-27 were similarly less active, and Av-14 the least active of the lead compounds. Lead compounds were evaluated for cytotoxicity in TZM-bl cells using a luminescence-based assay, CytotoxGlo, which measures luminescence produced by cleavage of the CytotoxGlo substrate by proteases released from damaged cells. We observed no significant cytotoxicity in Av-5, Av-14, and Av-26. However, Av-27 was cytotoxic at concentrations above 25 µM, with 100 µM treatment resulting in complete cell death (Figure 2b), and thus Av-27 was eliminated from further analyses. Our initial screening identified three promising HIV-1 antiviral compounds with negligible cytotoxicity, which we then further analyzed for their properties as potential HIV-1 antiviral therapies.

### 3.2. Evaluation of Antiviral Activity in Lead Compounds

The selected Avirulin compounds, Av-5, 14, and 26, were next evaluated for viral inhibition of HIV-1 BaL in the immortalized, T lymphocyte-derived cell line, PM1 (Figure 2a,b). These cells express the HIV target receptor, CD4, and both viral coreceptors, CCR5 and CXCR4, and retain many characteristics of primary human T cells, making them a relevant cell line for studying HIV infection in lymphocytes [29]. Infected PM1s were treated with Avirulins in culture for six days, to evaluate inhibition of an initial viral infection as well as its long-term potency after multiple rounds of viral replication. HIV-1 inhibition was determined by assessing virion release through the presence of the HIV-1 capsid protein p24 in cell supernatants (Figure 2b). We observed similar antiviral activity of Avirulin lead compounds in cells infected with BaL at day three post infection, but Av-5 was most effective after six days of infection. Av-5 50 µM and 10 µM treated cells had significantly less cell death at day six post infection compared to vehicle treated infected PM1s (*p* < 0.05). Based on these results, we chose Av-5 as a representative Avirulin compound for further analysis in primary cells and physiological fluids.

To confirm the activity of Avirulins in primary cells, we infected human peripheral blood mononuclear cells (PBMCs) with HIV-1 BaL and treated them with Av-5 following the protocol described for PM1 infections (Figure 2e,f). Av-5 effectively inhibited HIV-1 BaL in PBMCs (Figure 2), though some cytotoxicity was also observed at 50 µM, as there were significantly less live cells present (*p* < 0.05). The high antiviral activity and minimal cytotoxicity of Av-5 in primary cells underscores their potential as candidates for development into novel antiviral therapies.

### 3.3. HIV-1 Reverse Transcriptase Inhibition of Avirulins

We next investigated the mechanism of Avirulin HIV-1 inhibition by evaluating Avirulins against three HIV drug targets, HIV-1 reverse transcriptase (RT), integrase, and protease, using commercially available in vitro assays. Avirulins Av-5, Av-14, and Av-26 were not active against HIV-1 integrase or HIV-1 protease [33] (data not intended for publication). However, Avirulins exhibited dose-dependent inhibition of HIV-1 RT (Figure 3a). Av-5 exhibited the greatest inhibition of RT, while Av-14, was the least effective of the lead compounds.

We obtained RT-inhibitor resistant strains from the NIH AIDS Reagent Program to understand further Avirulins’ antiviral mechanism and evaluate activity against current RT-resistant strains (Figure 3b). There are currently two classes of RT-inhibitors; competitive nucleoside RT inhibitors (NRTIs), analogues of nucleotides bound at the RT active site; and, nonnucleoside RT-inhibitors, noncompetitive inhibitors that bind at the nonnucleoside drug binding site. We tested Av-5 against two NRTI-resistant strains, HIV-1_74V_/MT-2 and AZT resistant 215Y, two NNRTI-resistant strains, Nevirapine-resistant HIV-1 (N119), and HIV-1IIIB A17 variant, as well as one strain resistant to both NRTI and NNRTI, HIV-1_RTMDR1_/MT-2, using a TZM-bl luciferase reporter assay (Figure 3b). NRTI-resistant strains containing the drug resistant mutations 215Y, 74V, and 41L, did not reduce drug activity of Av-5, and were more sensitive to Av-5 than HIV-1 BaL, though this is likely due to diminished viral fitness caused by RT mutations. Av-5 was not effective against NNRTI-resistant strains containing the K103 and 181C mutations and the HIV-1IIIB A17 variant was completely resistant to Av-5. The multidrug resistant strain HIV-1_RTMDR1_/MT-2 had the NNRTI resistant mutation 106A, however this strain remained sensitive to Av-5. Based on our results of the in vitro RT-inhibition assay, and the sensitivity of RT-inhibitor resistant mutants to Av-5, it is highly likely that Av-5 inhibits HIV-1 RT through a noncompetitive mechanism, with binding likely reliant on K103 and 181C of the RT-nonnucleoside binding site.

### 3.4. Inhibition in Genetically Variant HIV-1 Clades HIV

HIV-1 RT has extremely low fidelity of replication, allowing for rapid mutation and development of drug resistance [34]. Therefore, as HIV-1 spread globally, it has become genetically diverse, and is separated by sequence into genetic subtypes, or clades. Neglecting HIV-1 clade diversity during drug development creates potential bias when these drugs are implemented at the global scale, as certain clades have distinct sensitivities to many widely used antiretroviral drugs. We therefore obtained clinical isolates of HIV-1 clades A, B, C, D, and F from the NIH AIDS Reagent Program for screening against Av-5. We first treated TZM-bl cells with virus containing PM1 supernatants for 24 h with or without various dilutions of Av-5 (Figure 4a). In TZM-bl cells, we determined Av-5 most effective against clades B, D, and F, with no significant difference compared to inhibition of BaL, while clades A and C were significantly less sensitive to Av-5, with IC_50_ of 5.23 and 4.15 µM, respectively (*p* < 0.05). We also infected PM1 cells with each clinical isolate and noticed no significant decrease in sensitivity to Av-5 at day three or day six of infection, when compared to laboratory strain BaL (Figure 4b,c) (significance considered *p* < 0.05, adjusted with Bonferroni correction for multiple comparisons). At day six post infection (Figure 4c), Av-5 was significantly more effective against clinical isolate of clade F at 0.4 µM (*p* = 0.01) Notably, the clade D isolate contained the R5 × 4 tropism, which can enter target lymphocytes using the cofactors CCR5 or CXCR4, rather than just CCR5, like the R5 viral tropism. We observed no difference in Av-5 sensitivity between these tropisms. Overall, Av-5 was effective against diverse clades, suggesting broad therapeutic potential.

### 3.5. Evaluation in the Female Reproductive Tract

In regions where women are at high risk of HIV-1 transmission, and condom use is low, topical microbicides are often considered the most effective preventatives [35]. However, even drugs with potent anti-HIV-1 activity can increase the risk of transmission if cytotoxicity damages the protective FRT mucosa. To screen for potential cytotoxicity in the FRT, we tested our lead compounds Av-5, Av-14, and Av-26 against cell lines representing the vaginal wall, ectocervix, and endocervix areas (VK2, Ect1, and End1, respectively), which maintain structural and metabolic features of these anatomical regions [36]. The CytotoxGlo assay was used to measure cytotoxicity of FRT cells in response to Avirulins, as described previously (Figure 5a–c). Avirulins were not significantly cytotoxic to FRT epithelial cells at concentrations up to 100 µM, when compared to vehicle control. Based on these results, we predict Avirulins could be used as topical inhibitors without damaging the FRT epithelium.

We next addressed whether Avirulins would maintain potency as topical antivirals in the complex and biologically active environment of the FRT, which contains factors that could potentially reduce drug potency [37]. We tested Avirulin antiviral activity against BaL in cervico–vaginal fluid (CVF) obtained from healthy premenopausal adult donors. We used CVF pooled from 5 different donors and clarified of mucus. We incubated the clarified CVF with Av-5 for 2 h at 37 °C to allow for interaction at physiological temperature. CVF from healthy women is inherently antiviral, as displayed in Figure 5d, and reported elsewhere [38,39,40].

We chose the concentration 5 µM Av-5, as this concentration inhibits BaL ~80% without the presence of CVF, making potential decreases in antiviral activity easier to detect. Av-5 antiviral activity was not diminished in the presence of CVF, and at 1:40 dilution of CVF, inhibition was significantly more effective when compared to inhibition without CVF (*p* < 0.05). Additionally, we confirmed CVF was not cytotoxic to TZM-bl cells compared to vehicle control (Figure 5e).

## 4. Discussion

To eradicate HIV, researchers must continue the development of novel antivirals to inhibit new infections and continue viral suppression in seropositive individuals. In this study, we identified a unique class of antiviral compounds, Avirulins. Of 32 compounds sharing the Avirulin phenoxyethyl–pyrazole–pyrrolidine scaffold, 24 compounds had measurable antiviral activity, and four compounds, Av-5, Av-14, Av-26, and Av-27 exhibited low micromolar antiretroviral activity. We assessed these compounds against current challenges in HIV-1 epidemiology; the rise of antiviral resistance and of male-to-female transmission. The chemical structure of Avirulins is distinct from other known antiretroviral drugs, making this class of compounds potentially valuable additions to the multidrug cocktails of ART. To our knowledge, no drug activity has been reported elsewhere for this class of compounds, though one group has reported unrelated activity of related pyrrolidine–pyrazole urea compounds with dehydrogenase inhibition activity [41]. Av-5, Av- 26, and Av-14 were not cytotoxic at concentrations as high as 100 µM. Av-5, the compound with the lowest IC_50_ and least cytotoxicity, also inhibited HIV-1 in PBMCs.

RT-inhibitors form the backbone of conventional ART regimens, but have a low genetic barrier to resistance, compared to other drug classes, such as protease inhibitors [42]. Considering this, discovery and development of novel RT-inhibitors may become increasingly important to treatment-experienced patients from ART failure. Therefore, the predicted RT-inhibition drug mechanism of Avirulins makes these compounds potentially beneficial components of antiviral therapies. Additionally, the low cytotoxicity of Avirulins makes them valuable candidate RT-inhibitors, as currently approved drugs of this class often have higher toxicity compared to other drug classes used in ART [43].

We predict Avirulins are NNRTIs; NRTI-resistant strains of HIV-1 were all susceptible to Av-5, while certain NNRTI-resistant strains were resistant. The NNRTI Delavirdine shares some chemical similarity to the high activity Avirulin compounds, with a similar molecular size and charged chemical substitute groups [44]. Of the NNRTI-resistant strains tested, the K103N strain was fully resistant to Av-5. K103N, grants HIV-1 complete resistance to the NNRTI Nevirapine and the second generation NNRTI, Efavirenz [43]. Avirulins were also less effective against 215Y and Y181C NNRTI-resistance mutants, though encouragingly, Av-5 exhibited only 2-fold decreased activity against the NNRTI resistant strain containing Y181C. This same mutation grants the virus full resistance to Nevirapine, and intermediate resistance to many other FDA approved NNRTIs, including Efavirenz, and Etravirine, an NNRTI considered less susceptible to resistance than other NNRTIs [43]. Additionally, seropositive patients with the NNRTI-resistant mutation 106A could potentially benefit from development of Avirulins, as this mutation did not affect antiviral potency.

Currently, research on antiretroviral resistance relies heavily on the predominant clade in North America, clade B [45]. However, clade C and A are more common in Africa, where the majority of new infections occur [46]. Clades A, B, and C are most common globally, with clade A located in central and Eastern Africa, clade B predominating in Western Europe and the Americas, and clade C in southern Africa and India [47]. Clade D is most commonly found in North Africa and the middle east, while Clade F is most often found in Southeast Asia and Eastern Europe [46,47]. Antiviral drug sensitivity has been known to differ between clades. For instance, some groups have reported clade C as less responsive to ART than other clades [48,49]. Av-5 inhibited infection in five HIV-1 clades A, B, C, D, and F, as well as R5 and R5 × 4 viral tropisms, suggesting Av-5 activity will not be affected by typical genetic variances. Of the 5 viral clades tested, clade F was most sensitive to Av-5. This is advantageous, as clade F was the most frequent in Southeast Asia, an area currently experiencing high infection rates [50]. Further studies of Avirulins should include more diverse viral targets, including HIV type 2 (HIV-2), as this type can be inherently resistant to NNRTIs [51].

Avirulins also have potential as topical microbicides for the FRT. Av-5, 14, and 26 were not cytotoxic to cell lines derived from the vaginal wall, ectocervix, or endocervix, even at high micromolar concentrations. As drugs with even moderate toxicity in the FRT can increase risk of infection, the negligible cytotoxicity of Avirulins in FRT cell lines is promising. In addition, antiviral activity was not diminished against biologically active CVF, suggesting their potential in the complex environment of the FRT mucosa. Further studies are needed to determine activity in other environments, such as oral and anal mucosa.

Based on their potency in diverse HIV-1 strains and low cytotoxicity, we predict Avirulin compounds will be valuable additions to the HIV-1 drug discovery pipeline. Given the low-to-mid micromolar concentrations needed for inhibition using this class of compounds, refinement would be required prior to clinical application. Nevertheless, the Avirulin chemical scaffold could likely serve as a beneficial component of ART, or as a topical antimicrobial preventative for the FRT.

## Figures and Tables

**Figure 1 viruses-11-00408-f001:**
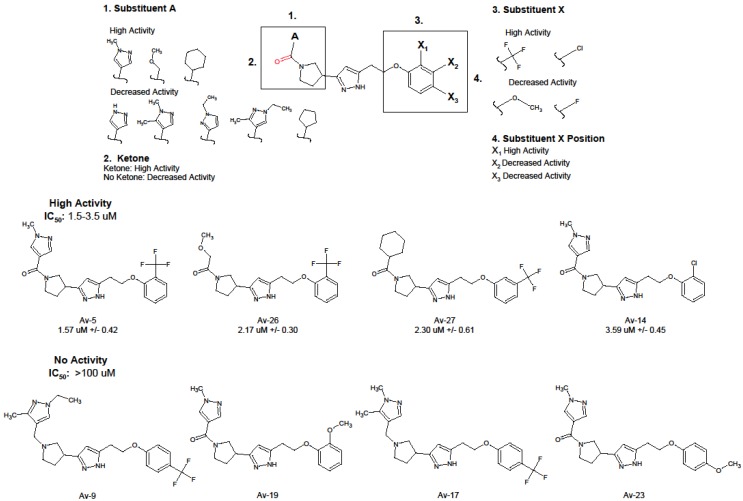
Structural and antiviral properties of Avirulins. Thirty compounds with structural similarity to the initial active compound were selected using a chemical database search program, SciFinder, and purchased from Asinex Corporation. Human immunodeficiency virus type 1 (HIV-1) antiviral inhibition of each compound was determined by a luciferase-based reporter assay measuring BaL infection of the HeLa derived cell line TZM-bl. TZM-bl contain the luciferase gene controlled by the HIV-1 Tat promoter and produce luminescence in response to infection. Inhibition was calculated by comparison of luminescence to the equivalent DMSO vehicle control. All 31 compounds were initially screened in triplicate at 50 µM for antiviral activity. Avirulins with detectable inhibition were further screened at lower concentrations for comparison of antiviral activity. Compounds with no inhibition at 50 µM were excluded from further evaluation. Trends in structural modifications to the Avirulin scaffold and the resulting change in antiviral activity and structures of four highly active Avirulins chosen as lead compounds for further analysis, and a selection of Avirulins with no antiviral activity are reported for structural comparison.

**Figure 2 viruses-11-00408-f002:**
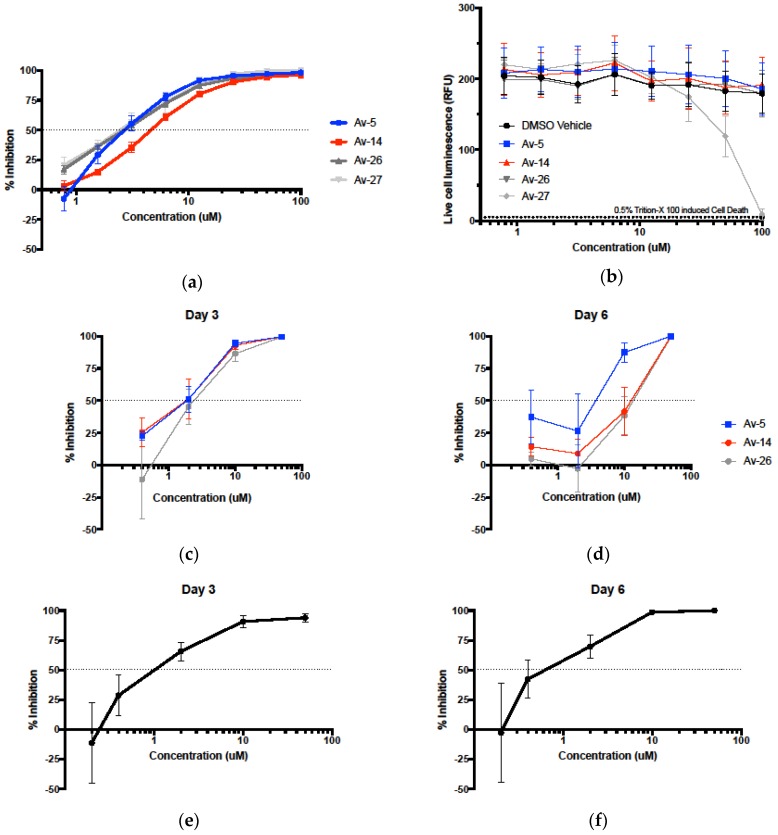
Antiviral activity and cytotoxicity of Avirulin lead compounds. (**a**) Lead compounds Av-5, Av-14, Av-26, and Av-27 were evaluated using a TZM-bl luciferase assay at a concentration range of 0.78 to 100 µM, for determination of IC_50_ and IC_90_ (*n* = 8). (**b**) Cytotoxicity of lead Avirulins towards TZM-bl was determined using a luminescence-based cytotoxicity assay, CytotoxGlo. Proteases released from dead cells cleaved a luminescent substrate, producing luminescence. Luminescence was normalized to total cell number by complete cell lysis with detergent. Av-27 was excluded at this point due to cytotoxicity. Avirulin lead compounds with low cytotoxicity, Av-5, Av-14, and Av-26 were used to treat BaL infected PM1 cells, an immortalized line of T lymphocytes (**c**,**d**). Based on its high activity and low cytotoxicity, Av-5 was selected to treat peripheral blood mononuclear cells (PBMCs) (**e**,**f**) isolated from healthy human subjects. PM1s and PBMCs were treated with Avirulins, then incubated with BaL for 1.5 h, followed by drug retreatment and incubation. Cell supernatants were collected on days 3 and 6 post infection, and cells were resuspended in fresh media with new drug on day 3 following infection to evaluate prolonged inhibition. Viral inhibition was determined by comparison of ng/million cells of the viral protein p24, detected by enzyme linked immunosorbent assays (ELISA) in cell supernatant of DMSO vehicle vs. drug treatment. PM1s: *n* = 4, PBMCs: *n* = 4, error = standard error of the mean (SEM).

**Figure 3 viruses-11-00408-f003:**
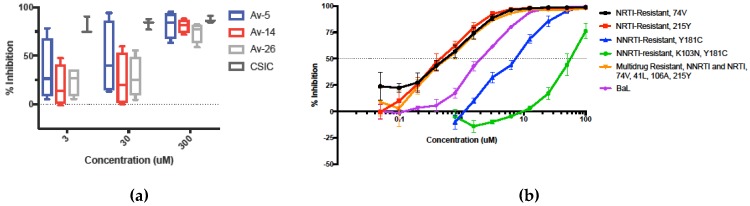
Reverse Transcriptase Inhibition Activity of Avirulin Lead Compounds. For determination of the viral target of Avirulins, commercially available assays for detection of HIV-1 integrase, protease, and reverse transcriptase (RT)-inhibition, were performed with lead compounds Av-5, Av-14, and Av-26. Avirulins showed no HIV-1 integrase inhibition or protease inhibition, but dose dependent inhibition of HIV-1 RT was found using a commercially available RT-inhibition assay (Roche) (**a**). The nonnucleoside reverse transcriptase inhibitor (NNRTI), 5-chloro-3-(phenylsulfonyl)indole-2-carboxamide (CSIC), was used as a positive control, and RT-inhibition was calculated by comparison with DMSO vehicle equivalent, with experiments performed in duplicate, *n* = 4, error = SEM. (**b**) To evaluate Avirulin antiviral activity in drug resistant strains of HIV-1, Av-5, the lead compound with greatest RT-inhibition and lowest IC_50_, was used to inhibit common RT-inhibitor resistant strains, two nucleoside RT-inhibitors, two nonnucleoside reverse transcriptase inhibitors, and one multidrug resistant strain, using a luciferase TZM-bl reporter assay. Inhibition was calculated by comparison to DMSO vehicle control. *n* = 4, error = SEM. Resistant strains were clinical isolates or inhibitor desensitized laboratory strains acquired through the NIH AIDS Reagent Program and expanded in PM1 cell culture. PM1 cell supernatants were used for infections.

**Figure 4 viruses-11-00408-f004:**
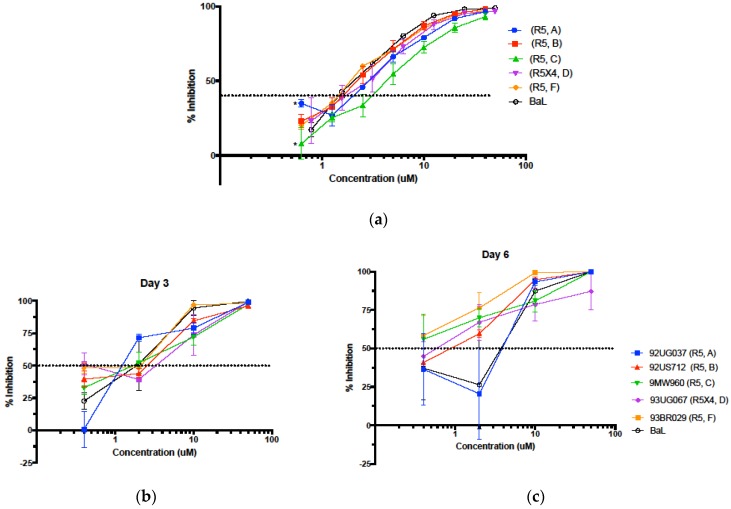
Avirulins are active against several HIV-1 clades and viral tropisms. Av-5 was used to inhibit HIV-1 clinical isolates from a wide range of clades using a TZM-bl luciferase reporter assay. Clinical isolates were obtained from the NIH AIDS Reagent Program and grown in PM1 cells, then viral supernatants were used to infect TZM-bl for 24 h (**a**). Percent inhibition was calculated by comparison to DMSO vehicle control (a). *n* = 3, error = SEM, inhibition against each clade was compared to HIV-1 BaL control, * = *p* < 0.05. PM1 cells were treated with Av-5 or DMSO, incubated for 1.5 h with clinical HIV-1 isolates, then PM1s were cultured for six days in the presence of Av-5 or DMSO, with retreatment on day three (**b**,**c**). Percent inhibition was determined by comparison of ng p24/million cells, detected by ELISA.

**Figure 5 viruses-11-00408-f005:**
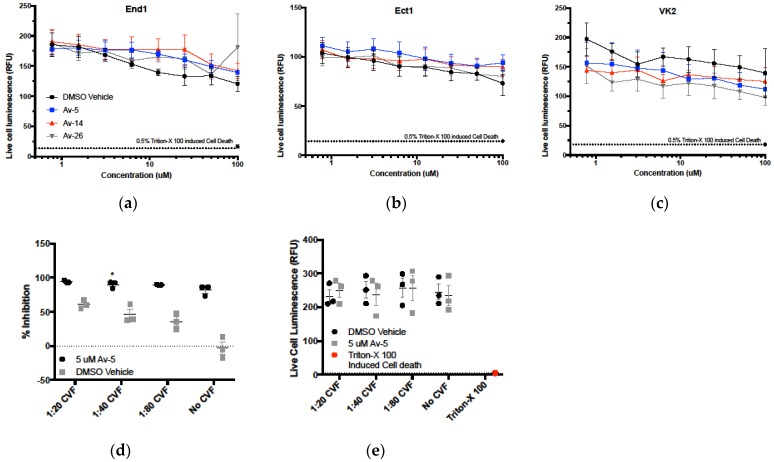
Cytotoxicity and antiviral activity of Avirulins in female reproductive tract epithelium. (**a**–**c**) Immortalized epithelial cell lines derived from the vaginal wall, ectocervix, and endocervix (VK2, Ect1, and End1, respectively) were treated with Avirulin lead compounds Av-5, Av-14, and Av-26 at a range of concentrations from 0.78–100 µM for 24 h. Cytotoxicity was measured using a commercially available luminescence-based cytotoxicity assay, CytotoxGlo. Cytotoxicity of treatments was compared to DMSO vehicle control, with the detergent Triton-X 100 for complete cell death. (*n* = 4, error = SEM). (**d**) Av-5 was incubated with clarified vaginal fluid at 37 °C for 2 h, then diluted with culture media and applied to TZM-bl. Cells were infected with BaL for 24 h, then inhibition was determined using the luciferase reporter assay. Cervico–vaginal fluid from healthy women is inherently antiviral, as shown in the vehicle control. *n* = 3, error = SEM * = *p* > 0.05. (**e**) Cytotoxicity of vaginal fluids and 5 µM Av-5 on TZM-bl measured using CytotoxGlo assay. Cervico–vaginal fluid (CVF) dilutions with Av-5 or equivalent DMSO were incubated at 37 °C for 2 h, then applied to TZM-bl. Cell death at 24 h was determined by subtraction of total cell luminescence from dead cell luminescence. *n* = 3, error = SEM.

**Table 1 viruses-11-00408-t001:** Avirulin lead compounds.

Compound	IUPAC	IC_50_ (uM) ^1^	IC_90_ (uM) ^1^
Av-5	3-[1-(1-methyl-1*H*-pyrazole-4-carbonyl)pyrrolidin-3-yl]-5-{2-[2-(trifluoromethyl)phenoxy]ethyl}-1*H*-pyrazole	1.57 ± 0.42	8.41 ± 1.26
Av-26	2-methoxy-1-[3-(5-{2-[2-(trifluoromethyl)phenoxy]ethyl}-1*H*-pyrazol-3-yl)pyrrolidin-1-yl]ethan-1-one	2.17 ± 0.30	15.9 ± 1.21
Av-27	3-(1-cyclohexanecarbonylpyrrolidin-3-yl)-5-{2-[3-(trifluoromethyl)phenoxy]ethyl}-1*H*-pyrazole	2.30 ± 0.61	18.2 ± 1.61
Av-14	5-[2-(2-chlorophenoxy)ethyl]-3-[1-(1-methyl-1*H*-pyrazole-4-carbonyl)pyrrolidin-3-yl]-1*H*-pyrazole	3.59 ± 0.45	18.65 ± 1.18

^1^ TZM-bl cells were infected with HIV-1 BaL treated with 0.78–100 µM drug treatment. Fifty percent inhibitory concentrations (IC_50_), IC_90_, and error was calculated by nonlinear fit using GraphPad Prism. Experiments were repeated *n* = 7 for Av-5, Av-14, and Av-26 and *n* = 3 for Av-27.

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
