# Peer review of "Avirulins, a Novel Class of HIV-1 Reverse Transcriptase Inhibitors Effective in the Female Reproductive Tract Mucosa"

_viruses, 2019, doi:10.3390/v11050408_

Round 1

Reviewer 1 Report

See attached file

Author Response

We thank the reviewers for their time and insightful comments reviewing this manuscript. Below are our point-by-point responses to all reviewers’ comments.

Response to Reviewer 1’s comments:

Line 16: “Asinex Corp.” What is Asinex Corp.?

A more in-depth description of Asinex Corporation has been provided in the abstract. 

Line 22: “nonnucleoside” Which one?

The specific resistant mutation has now been included in the abstract.  

Lines 26-27: “human cervicovaginal fluid, suggesting retained potency in the milieu of the FRT” What about the seminal fluid and oral/anal mucosa?

Avirulins were not tested for activity in oral/anal mucosa or seminal fluid due to resource limitations, but this point has been included as a possible future direction in the discussion section.  

Lines 47-48: “preventative treatments such as vaccines” There are no vaccines against HIV

“The development of” has been included, to help clarify the present lack of vaccines.

Lines 67-68: “currently six approved classes of antiretroviral drugs” Which six?

The drug classes have been included at the end of this sentence for clarity.

Line 86: “Asinex Corporation” What is Asinex Corp.?

A more thorough explanation of Asinex Corp. has been included in the introduction

Line 92: “not cytotoxic” Indicate concentrations to which they were not cytotoxic

The concentration has been included.

Line 98: “TZM-bl” Provide more details on TZM-bl cells

“The immortalized cell lines used for HIV-1 infections” has been included to specify the function of this cell line. 

Line 183: “CSIC” What is CSIC?

The full chemical name and the corresponding citation of CSIC has been added to the methods. 

Line 218: “…purchased from Asinex corp” Purchased at what cost?Can avirulins be considered NNRTIs? Are they active against HIV-2? What is SAR of aviruline?

Good questions - we predicted Avirulins are NNRTIs based on their activity against specific NNRTI resistant mutants. Future structural studies would be useful to confirm this, which would also help define SAR. Avirulins were not tested against HIV-2, but a sentence suggesting this experiment as a further direction has been included in the discussion. 

Line 262: “Lead” Chosen as “lead” based, solely on potency?

The 4 ‘lead’ compounds (Av-5, Av-26, Av-27, and Av-14) were moved forward in analyses based on their high potency. However, describing only Av-5, Av-26, and Av-14 “lead” compounds in section 3.2 may have been confusing, so this word has been removed. 

Line 399: “”we confirmed determined” Rephrase!

This phrase has been corrected as suggested.

Line 423:         “suggests”Only suggestive

                        “as”Replace by “;”

“Suggests” has been replaced with “We propose”. 

Line 453:  “in” Replace by “against”

“in” has been replaced here with “against”. 

Reviewer 2 Report

The manuscript entitled: "Avirulins, a Novel Class of HIV-1 Reverse Transcriptase Inhibitors effective in the Female Reproductive Tract Mucosa" by dr. Cherne and colleaugues is well written and suitable for publication. They screened a library of  new compounds (Avirulins) for anti-HIV  activity. Based on their  promising results, they provide new insights for development of next-generation treatments and preventatives that target HIV-1.

Author Response

Response to Reviewer 2’s comments:

(reviewer #2 did not have specific comments to address)

Reviewer 3 Report

In this manuscript the authors describe the discovery of a new class of compounds referred to as “Avirulins” that are effective HIV inhibitors.  The compounds directly inhibit reverse transcriptase (RT) and resemble NNRTIs based on the proposed mechanism of action.  The compounds were discovered after a screen or a library from “Asinex”.  One antiviral compound was discovered from this screen.  The structure of this compound was used to search for other available compounds with related structure.  Several compounds were identified and screened and 4 were pursued based on their high antiviral activity.  The compounds demonstrated modest activity in comparison to current NNRTIs (IC50 1.5-3.6 uM).  For example, CSIC use in Fig. 3 appears considerably more potent than any of the Avirulins and other.  Other NNRTIs (e.g. EFV, RPV) typically have low nM IC50s.  However, Avirulins, with the exception of AV27, showed low toxicity and overall the results suggest that they could be a good starting point for the development of a novel RT inhibitor class.  Experiments appear to have been carefully performed and appropriate, however, several points need clarification:

-          The reason for choosing the Asinex library is unclear, and the nature and derivation of that library is not described.

-          The structure of the original compound isolated from the Asinex library should be shown.

-          Table 1 should have a legend briefly describing the number of experiments and statistical analysis and type of experiments that were used.

-          In Fig. 2 several graphs have y axis that are unnecessarily extended in the negative direction.  It would be more effective to make the figure larger and remove the unnecessary areas. 

-          Fig. 3, the key is cutoff and unreadable in both A and B.  Boxed area and the line inside the box in 3A should be explained.

-          Fig. 4, the description of the p value is not clear.  It is described in the results section but should be made clear in the legend that it is with respect to HIVbal control.

-          Fig 5 is mislabeled as “Fig. 1”.  The font sizes on the graphs also vary between different panels and are small in some cases.

-          IC50 is written wrong on line 285 and in the legend to Fig. 2.

-          Line 53- put “MSM” in parentheses on line 53 before reference.

-          Line 89- delete the first “scaffold”

-          Line 139- 50 ml conical flask or tube?

-          Line 148- Remove the “)” after the “Corporation.”

-          Line 169- P24 is written as p24.

-          Line 181- “measure with the Roche”

-          Line 271- HIV p24 is a capsid not an envelope protein.

-          Line 326- “HIV-1RTMF/MT-2, HIV-1RTMDR1/MT-2” it is not clear that this is one strain that is resistant to both drugs.

-          Line 338- “is it has become”  change to “it has become”

Author Response

Response to Reviewer 3’s comments:

The reason for choosing the Asinex library is unclear, and the nature and derivation of that library is not described.

A more thorough description of the Asinex library has been included in the introduction.

The structure of the original compound isolated from the Asinex library should be shown.

The original compound from the Asinex library is Av-5, but this was not clear in the text. It has been added to the results section.- “From this library, we discovered one antiviral compound which inhibited 50% of viral infection at 1.5 µM, which was referred to as Av-5.”

Table 1 should have a legend briefly describing the number of experiments and statistical analysis and type of experiments that were used.

As suggested, a footnote/legend has been added to the table, which includes this information. 

In Fig. 2 several graphs have y axis that are unnecessarily extended in the negative direction.  It would be more effective to make the figure larger and remove the unnecessary areas. 

As suggested, Y axes have been reduced. 

Fig. 3, the key is cutoff and unreadable in both A and B.  Boxed area and the line inside the box in 3A should be explained. 

These figures have been corrected- The x-axis of 3A has been reformatted. 

Fig. 4, the description of the p value is not clear.  It is described in the results section but should be made clear in the legend that it is with respect to HIVbal control.

A description of the statistical analysis used has been included. 

Fig 5 is mislabeled as “Fig. 1”.  The font sizes on the graphs also vary between different panels and are small in some cases. 

5D and 5E have been reformatted.

IC50 is written wrong on line 285 and in the legend to Fig. 2.

Corrected.

Line 53- put “MSM” in parentheses on line 53 before reference.

“MSM” has been included. 

Line 89- delete the first “scaffold” 

The extra word has been deleted. 

Line 139- 50 ml conical flask or tube?

“Tube” has been included for clarification. 

Line 148- Remove the “)” after the “Corporation.”

Corrected. 

Line 169- P24 is written as p24.

Corrected. 

Line 181- “measure with theRoche”

“With the” has been added. 

Line 271- HIV p24 is a capsid not an envelope protein.

Corrected. 

Line 326- “HIV-1RTMF/MT-2, HIV-1RTMDR1/MT-2” it is not clear that this is one strain that is resistant to both drugs.

The strain was accidentally included twice; this has been corrected. 

Line 338- “is it has become”  change to “it has become”

Corrected.